# Unlocking the Potential of Octocoral-Derived Secondary Metabolites against Neutrophilic Inflammatory Response

**DOI:** 10.3390/md21080456

**Published:** 2023-08-18

**Authors:** Ngoc Bao An Nguyen, Mohamed El-Shazly, Po-Jen Chen, Bo-Rong Peng, Lo-Yun Chen, Tsong-Long Hwang, Kuei-Hung Lai

**Affiliations:** 1Graduate Institute of Pharmacognosy, College of Pharmacy, Taipei Medical University, Taipei 11031, Taiwan; m303110001@tmu.edu.tw (N.B.A.N.); peng_br@tmu.edu.tw (B.-R.P.); m303110004@tmu.edu.tw (L.-Y.C.); 2Department of Pharmacognosy, Faculty of Pharmacy, Ain-Shams University, Organization of African Unity Street, Abassia, Cairo 11566, Egypt; mohamed.elshazly@pharma.asu.edu.eg; 3Department of Medical Research, E-Da Hospital, I-Shou University, Kaohsiung 82445, Taiwan; ed113510@edah.org.tw; 4Research Center for Chinese Herbal Medicine, College of Human Ecology, Chang Gung University of Science and Technology, Taoyuan 33303, Taiwan; 5Graduate Institute of Health Industry Technology, College of Human Ecology, Chang Gung University of Science and Technology, Taoyuan 33303, Taiwan; 6Department of Anaesthesiology, Chang Gung Memorial Hospital, Taoyuan 33305, Taiwan; 7Graduate Institute of Natural Products, College of Medicine, Chang Gung University, Taoyuan 33302, Taiwan; 8Department of Chemical Engineering, Ming Chi University of Technology, New Taipei City 24301, Taiwan; 9PhD Program in Clinical Drug Development of Herbal Medicine, College of Pharmacy, Taipei Medical University, Taipei 11031, Taiwan; 10Traditional Herbal Medicine Research Center, Taipei Medical University Hospital, Taipei 11031, Taiwan

**Keywords:** neutrophilic inflammation, octocoral, secondary metabolites, drug leads

## Abstract

Inflammation is a critical defense mechanism that is utilized by the body to protect itself against pathogens and other noxious invaders. However, if the inflammatory response becomes exaggerated or uncontrollable, its original protective role is not only demolished but it also becomes detrimental to the affected tissues or even to the entire body. Thus, regulating the inflammatory process is crucial to ensure that it is resolved promptly to prevent any subsequent damage. The role of neutrophils in inflammation has been highlighted in recent decades by a plethora of studies focusing on neutrophilic inflammatory diseases as well as the mechanisms to regulate the activity of neutrophils during the overwhelmed inflammatory process. As natural products have demonstrated promising effects in a wide range of pharmacological activities, they have been investigated for the discovery of new anti-inflammatory therapeutics to overcome the drawbacks of current synthetic agents. Octocorals have attracted scientists as a plentiful source of novel and intriguing marine scaffolds that exhibit many pharmacological activities, including anti-inflammatory effects. In this review, we aim to provide a summary of the neutrophilic anti-inflammatory properties of these marine organisms that were demonstrated in 46 studies from 1995 to the present (April 2023). We hope the present work offers a comprehensive overview of the anti-inflammatory potential of octocorals and encourages researchers to identify promising leads among numerous compounds isolated from octocorals over the past few decades to be further developed into anti-inflammatory therapeutic agents.

## 1. Introduction

Inflammation is an integral response of multicellular organisms that protects the hosts from external harmful factors, such as pathogens and physical and chemical irritants. It also helps in tissue recovery after injury [1]. It can be classified into acute or chronic inflammatory responses, depending on the duration of the process and cellular activities [2]. Acute inflammation is the first response of the defense system to the invasion of foreign stimuli, which involves a cascade of complicated events. The extravascular migration of immune cells, such as platelets, basophils, neutrophils, eosinophils, mast cells, and macrophages, is one of the characterized features of acute inflammation, which aims to remove inflammatory irritants and facilitate the recovery of tissues [3,4]. The expected result of the acute phase is to eliminate either infectious or non-infectious agents and to restore the tissue to its initial state. If the exogenous stimuli cannot be removed entirely or the reactions of the acute phase are not sufficient to resolve the damage in the inflamed area, the inflammatory state can persist and proceed to the chronic phase [5,6]. Various studies have demonstrated the relationship between chronic inflammation and serious health problems, including type II diabetes, dyslipidemia, chronic kidney diseases, chronic prostatic diseases, cardiovascular diseases, and many types of cancer [7,8,9,10,11,12].

Neutrophils are the most abundant polymorphonuclear leukocytes, representing 50–70% of all white blood cells. They exhibit a significant role in the acute inflammatory response. Recently, the crucial role of neutrophils in various chronic inflammatory diseases has been investigated by many research groups. This investigation resulted in a wide range of studies focusing on the development of neutrophil-regulating agents as potential anti-inflammatory therapeutics [6,13,14,15,16,17,18]. Neutrophils act as the first-line defender of the immune system in the battle against noxious stimuli due to their rapid recruitment and various effective mechanisms of defense. The invasion of tissue with either infectious or non-infectious neutrophils triggers alarming signals, resulting in the recruitment and accumulation of neutrophils at the invaded sites. At these sites, neutrophils are activated and deploy their multiple defense mechanisms to protect the area from invaders (Figure 1). The offensive mechanisms of neutrophils include phagocytosis, respiratory burst, degranulation, and the formation of neutrophil extracellular traps [13,19,20].

At the invaded tissue, the activated neutrophils can act as phagocytes that engulf microorganisms, tissue debris, and dead cells in phagosomes similar to macrophages. However, there are some differences in the phagocytosis of neutrophils as compared to that of macrophages, especially in the maturation process of phagosomes. Unlike macrophages, neutrophils contain many types of preformed granules in their cytoplasm used for defense, including primary granules (known as azurophilic granules), secondary granules (referred to as specific granules), tertiary granules (another name for gelatinase granules), and secretory vesicles. For the maturation process, phagosomes require fusions with these granules to perform their degradative function against pathogens and foreign particles. Subsequently, the microorganisms and debris can be degraded and eliminated by granule proteins, and the products of oxygen burst [21].

In addition to phagocytosis, respiratory burst is another defensive method of neutrophils. Reactive oxygen species (ROS) are highly reactive derivatives of oxygen, which are produced by nicotinamide adenine dinucleotide phosphate (NADPH) oxidase complex via respiratory burst in activated neutrophils. Superoxide anions (O_2_^•−^) are the initial product of respiratory burst, which are converted into peroxide (H_2_O_2_) by superoxide dismutase and subsequently into hypochlorous acid (HOCl) by myeloperoxidase (MPO) or hydroxyl radical (OH^•^) through Fenton’s reaction in the presence of Fe^2+^. These ROS can be released into either the phagosome for the disposal of foreign particles or in the extracellular environment in case of too large microorganisms that cannot be phagocytosed. The products of respiratory bursts possess strong oxidizing properties that can oxidize cellular constituents and damage the DNA, resulting in the destruction of pathogens and the removal of foreign particles. Due to their highly reactive and toxic nature, the excessive and prolonged generation of ROS can also cause further damage to the inflamed tissues, resulting in a chronic inflammatory response [22,23,24].

If the microorganisms are too large to be digested by phagocytosis, degranulation and the formation of neutrophil extracellular traps can be deployed during the response. Azurophilic granules are the primary granules that contain a majority of the pro-inflammatory and antimicrobial proteins such as elastase, myeloperoxidase, cathepsin G, and defensins, which are released to the extracellular environment of the inflammatory site or into the phagosome in a process called degranulation. The remaining defensive mechanism of neutrophils is the extrusion of extracellular traps (NETosis) that are made of decondensed chromatin, elastase, myeloperoxidase, cathepsin G, and histones. The neutrophil extracellular traps (NETs) can prevent the spread of pathogens by capturing and digesting microorganisms. However, NETs can also be injurious to host tissues due to the proteolytic activity of proteases [13].

These offensive mechanisms of neutrophils are double-edged swords that not only protect the host from pathogens and other foreign particles but can also attack the surrounding area in the case of uncontrolled or overwhelmed activation, which results in a subsequent chronic disorder rather than a resolution of inflammation. The overproduction of ROS, especially superoxide anions as well as the activation of elastase enzyme, contribute to the damage of host cells if the activation of neutrophils is out of control. Superoxide anions are the primary reactive oxygen species produced by neutrophils as a defense mechanism of the host in response to contact with foreign stimuli. In addition to the destruction of the invading agents, they can also destroy the surrounding area as well, which may lead to further severe damage to inflamed tissues, especially in the case of imbalanced production [24]. Human neutrophil elastase is one of the proteases that are secreted during degranulation and is a crucial component of the NETs during combat against invading pathogens and sterile agents. The enzyme can degrade not only the foreign proteins in extracellular environments but also the host cell matrix, which can lead to a harmful effect in the case of over-activation. Numerous effects related to tissue damage have been linked to the imbalanced activity of elastase, making it a therapeutic target for various health problems, including chronic obstructive pulmonary disease [25], bronchiectasis [26], colorectal cancer [27], leukemia [28], cystic fibrosis [29], pulmonary arterial hypertension [30], and COVID-19-associated acute respiratory distress syndrome [31]. Due to the pivotal role of superoxide anions and elastase in the pathogenesis of inflammation-induced diseases, their production was selected to evaluate the anti-inflammatory effect of natural products.

Currently, anti-inflammatory drugs are produced from synthetic sources, including nonsteroidal and steroidal drugs. Among them, several neutrophil-targeting anti-inflammatory agents have been used in the clinical stage, such as colchicine, secukinumab, ixekizumab, brodalumab, reparixin, danirixin, sivelestat, and nafamostat. These conventional therapeutic agents exhibit potent effects in the treatment of various inflammation-related diseases, but their adverse effects cannot be ignored. Therefore, there is an increasing number of research studies focusing on the exploration of safer and more effective anti-inflammatory therapeutic agents. Nature was and will be the most valuable and sustainable source for drug discovery. The prolific source of novel lead compounds derived from natural products has been demonstrated via a huge range of studies [32,33,34]. In recent years, the emergence of marine drug exploration has inspired more research on evaluating different marine organisms, including soft corals, due to their diverse chemical constituents and promising biological effects [35,36,37,38,39,40,41,42,43,44,45,46,47,48,49,50,51,52]. Soft coral (Figure 2) is the common name of the marine animals classified under the subclass Octocorallia, class Anthozoa of the phylum Cnidaria [53]. They are widely distributed in tropical shallow water or the deep sea [54]. These marine organisms are named “octocorallia” as they possess eight pinnate tentacles on the oral opening of their polyp tubes, which are used as a tool for food capture. Unlike hard corals, more than 60% of the octocoral body is occupied by fleshy parts. Therefore, their defensive mechanism against potential predators mainly relies on the chemical composition contained in the soft tissues [55,56]. The use of soft corals was recorded in ancient literature as a therapeutic ingredient for the treatment of diarrhea, gastrointestinal bleeding, and neurasthenia [57]. In modern times, a great quantity of studies has been conducted in regard to the chemical constituents of soft corals and their pharmacological potentials. Terpenoids and steroids are reported as chemical compositions commonly discovered in these organisms. Secondary metabolites derived from different species of soft corals have shown a diverse range of bioactivities, including cytotoxic, antimicrobial, antimalarial, antifouling, antidiabetic, anxiolytic, antileishmanial, anti-acne, analgesic, antiviral, and anti-inflammatory effects [35,36,37,38,39,45,49,50,51,52,58,59,60,61,62]. The promising potential for the pharmacological effects of soft corals has leveraged them to be hot spots in the race for drug discovery.

Aquaculture of soft corals has been conducted since the late 1950s for the purposes of commercialization and preservation. Many techniques, such as coral gardening, micro-fragmentation, larval enhancement, and direct transplantation, have been developed to meet the demand for the mass production of soft corals [38]. Ex situ and in situ are current approaches to coral cultivation. Whereas in situ practice relies on natural environment for the propagation of soft corals, the ex situ method produces these marine organisms in controlled conditions. Although an ex situ approach is more costly and requires more advanced skills than in situ, it allows the optimization of aquaculture conditions and the elimination of environmental variability so as to facilitate and enhance the biomass and metabolite production of cultivated soft corals. Moreover, there is no interference in the growth of the animals concerning exposure to deleterious factors, such as parasites, competitors, predators, and other hazards [63]. These advantages make ex situ practice a suitable and favorable method of aquaculture to serve the drug discovery journey in marine organisms, particularly soft corals.

Although there are many reviews on these marine organisms, their inhibitory effects targeting neutrophil-mediated inflammation have not been summarized. Therefore, in this review, we aimed to provide a comprehensive summary of marine soft corals’ potential as a plentiful resource of neutrophil-targeting anti-inflammatory agents. Superoxide anion generation and elastase release in activated neutrophils were the most common methods used for the evaluation of the anti-inflammatory potential of these marine secondary metabolites.

A total of 299 compounds isolated from different species of the subclass Octocorallia were screened for their anti-inflammatory potential using the aforementioned in vitro tests. Among them, 97 isolates (Table 1) were considered to exhibit significant inhibitory effects on superoxide anion generation and elastase release, with IC_50_ equal to or less than 20 μM. In the current review, the secondary metabolites possessing significant neutrophil-targeting anti-inflammatory effects are classified into sesquiterpenes, diterpenes, biscembranes, steroids, and some miscellaneous compounds. Steroids are the most abundant population with a total of 48 compounds, occupying 49.5% of bioactive agents isolated from octocorals. Ranked in second place, diterpenes comprise 36 derivatives that are classified into 10 subtypes. The remaining 13.4% of 97 potent isolates include 6 sesquiterpenes, 6 biscembranes, 2 α-tocopherol derivatives, a nitrogen-containing compound, and an allenic norterpenoid ketone. The literature investigation was conducted using various scientific databases including PubMed, Google Scholar, ScienceDirect, ResearchGate, and Reaxys. Different keywords, such as “neutrophil”, “elastase release”, “superoxide anion”, “gorgonian”, “soft corals”, “octocoral”, and “secondary metabolites”, were used to retrieve original papers studying the target topic.

## 2. Soft Corals—The Source of Anti-Inflammatory Lead Compounds

### 2.1. Sesquiterpenes and Derivatives

Sesquiterpenes are a subclass of terpenoids, comprising three units of isoprene in the structure, or having a C15 skeleton with the molecular formula of C_15_H_24_. They can be acyclic or cyclic with various types of rings making their structures interesting from chemical and biological perspectives [64].

Several chemical studies conducted on *Rumphella antipathies* led to the isolation of bioactive sesquiterpenes, including clovan-2,9-dione (**1**), antipacid B (**2**), and rumphellolide L (**3**) (Figure 3). Clovan-2,9-dione, which was previously described as a synthetic compound, was isolated from a marine natural source as a natural clovane-type sesquiterpenoid for the first time. Antipacid B is a caryophyllane-related sesquiterpenoid possessing a novel bicyclo[5.2.0] carbon core skeleton. Both compounds significantly inhibited superoxide anion generation with IC_50_ values of 11.22 μM and 2.72 μg/mL. They also moderately inhibited elastase release with the IC_50_ values of 23.53 μM and 6.73 μg/mL. Rumphellolide L, a dehydrated product derived from the esterification reaction of the novel sesquiterpene, antipacid A, and a known sesquiterpene, clovane-2β,9α-diol, was also isolated from the same sample of the soft coral *Rumphella antipathies*. It showed a potent elastase release inhibitory effect with an IC_50_ value of 7.63 μM [65,66].

### 2.2. Diterpenes and Derivatives

Diterpenes are secondary metabolites that comprise four isoprene units possessing the basic molecular formula of C_20_H_32_. They often exist under highly oxygenated forms with acyclic or cyclic frameworks, which results in a wide range of different carbon skeletons [67]. Within the scope of the current literature-based investigation, four major subtypes of diterpenes, including briarane, cembrane, eunicellin, and xenicane, along with six minor diterpene classes, such as halimane, verticillane, C-2/C-20-cyclized cembranoid skeleton, norcembranoid, capsosane, and lobane, are included in this section.

#### 2.2.1. Briarane-Type Diterpenes

Briaranes are marine metabolites that are featured by a bicyclo[8.4.0]tetradecane skeleton fused with a γ-lactone ring (Figure 4). It was suggested that they are derived from the 3,8-cyclization of cembranes [60].

In biological activity assays, three new 8-hydroxybriarane diterpenoids were isolated from the Gorgonian corals *Junceella juncea*, junceols A–C (4–6). At 10 μg/mL, they inhibited superoxide anion generation by human neutrophils with 45.64%, 159.60%, and 124.14%, respectively [68].

A new briarane diterpenoid, briarenolide F (**7**), which was isolated from an octocoral *Briareum* sp., was suggested to be the first 6-hydroperoxybriarane derivative. This compound showed a significant inhibitory effect on the generation of superoxide anion by human neutrophils [69].

The chemical investigation of *Briareum* sp. yielded briarenolide J (**8**) which was identified as the first metabolite of a briarane-related natural product. It was found to possess a chlorine atom at C-12. It inhibited the generation of superoxide anions and the release of elastase by human neutrophils with IC_50_ values of 14.98 and 9.96 μM, respectively [70].

In in vitro anti-inflammatory activity assays, it was found that the new polyoxygenated briarane diterpenoid isolated from the octocoral *Briareum excavatum*, briarenol D (**9**), showed a selective inhibitory effect on the release of elastase with an IC_50_ value of 4.65 μM by human neutrophils [71].

In an in vitro anti-inflammatory activity assay, juncin Z (**10**), which was obtained from the gorgonian coral *Junceella fragilis*, showed a 25.56% inhibitory effect on the generation of superoxide anions by human neutrophils at a concentration of 10 μM [72].

#### 2.2.2. Cembrane-Type Diterpenes

Cembrane-type diterpenes (Figure 5) are macrocyclic metabolites that were suggested to be derivatives of geranylgeranyl pyrophosphate. They are structurally diverse with an extended family of subtypes due to the variations in functional groups and patterns of cyclization. Their basic 14-membered ring could be fused to lactones with 5- to 8-atom rings. Several moieties could be attached to the core frameworks, such as epoxide, ester, hydroxyl, peroxide, carboxyl, aldehyde, and ketone groups [58].

Lobocrassin B (**11**), which was isolated from the soft coral *Lobophytum crassum*, is a new cembrane metabolite that displayed significant inhibitory effects on the generation of superoxide anion and the release of elastase by human neutrophils, with IC_50_ values of 4.8 and 4.9 μg/mL, respectively. This compound was found to be a stereoisomer of the known cembranes, 14-deoxycrassin [73].

From the ethyl acetate extract of the soft coral *Sinularia arborea*, a new cembrane-type diterpenoid, arbolide C (**12**), was isolated. The new compound significantly inhibited the release of elastase enzyme with an IC_50_ value of 5.13 μg/mL [74].

A known cembrane isolated from the Formosan soft coral *Sarcophyton tortuosum*, emblide (**13**), inhibited elastase release to the extent of 29.2% at 10 μM [75].

A known metabolite, isosarcophytonolide D (**14**), was rediscovered from the cultured soft coral *Sarcophyton glaucum* and showed a 27.12% inhibitory effect on the elastase release at 10 μM [76].

A new cembrane analog, sinulerectol C (**15**), was obtained in the chemical exploration of a Dongsha Atoll soft coral *Sinularia erecta*. At 10 μM, the metabolite inhibited elastase release with an inhibitory rate of 33% [77].

The soft coral *Sinularia flexibilis* afforded a known cembranoid, 14-deoxycrassin (**16**), which inhibited superoxide anion generation and elastase release by human neutrophils with IC_50_ values of 10.8 and 11.0 μM, respectively [59].

A chemical investigation on Formosan soft coral *Klyxum flaccidum* led to the isolation of two potent inhibitors of elastase release. Flaccidodioxide (**17**), a new cembranoid, showed a 17.17% inhibitory effect at 10 μM, whereas the other known analog, 14-O-acetylsarcophytol B (**18**), demonstrated an elastase release inhibitory effect, with an IC_50_ value of 7.22 μM [78].

#### 2.2.3. Eunicellin-Type Diterpenes and Derivatives

Eunicellin-based diterpenes are characterized by a six-carbon ring fused to a ten-carbon ring with an ether bridge that connects C-2 and C-9 or C-4 and C-7 in the latter ring. In some cases, the bond between C-6 and C-7 in the ten-membered ring moiety can be broken to form 6,7-secoeunicellin derivatives [79] (Figure 6).

In a chemical exploration of an Indonesian octocoral identified as *Cladiella* sp., a new eunicellin-type diterpenoid that possessed a 2-hydroxybutyroxy group at C-4 was discovered and designated as cladielloide B (**19**). The metabolite showed a significant inhibitory effect on superoxide anion generation and elastase release with IC_50_ values of 5.9 and 6.5 μg/mL, respectively [80].

A chemical examination of the Formosan soft coral *Klyxum molle* resulted in the isolation of the first eunicellin-based metabolite with a phenylacetate moiety at C-6, klymollin M (**20**). The distinct substituent might contribute to the most potent anti-inflammatory activities of **20** toward the generation of superoxide anions (IC_50_ = 3.13 ± 0.39 μM) and elastase release (IC_50_ = 2.92 ± 0.27 μM) when compared to the eunicellines that were derived from the same organic extract of the soft coral [81].

A series of new eunicellin-type diterpenes was discovered in the soft coral *Cladiella krempfi*. Several of them were found to be effective anti-inflammatory agents. Krempfielin K (**21**), which possesses a rare eunicellin skeleton with a highly oxygenated pattern at C-2, C-3, C-6, C-7, C-8, C-9, and C-12, showed a 45.51% inhibitory effect on the release of elastase at a concentration of 10 µM. At the same concentration, krempfielin M (**22**) also significantly inhibited the elastase release (27.30 ± 5.42% inhibition). Krempfielin N (**23**) showed the most potent activity among the isolates, displaying an up to 73.86% inhibitory effect on elastase release at 10 µM, with an IC_50_ value of 4.94 µM. The remaining active eunicelline in this series was krempfielin P (**24**), which inhibited not only elastase release but also the generation of superoxide anion with 35.54% and 23.32% inhibition at 10 μM, respectively [82,83].

A formerly reported eunicellin metabolite, sclerophytin B (**25**), was rediscovered in the chemical exploration of an octocoral of *Cladiella* sp. The compound showed 28.12% inhibitory effects on human neutrophils in terms of the generation of superoxide anions at a concentration of 10 μM [84].

An NMR-guided chemical examination of the octocoral *Cladiella* sp. afforded the first secoeunicellin possessing two tetrahydrofuran moieties, which was named cladieunicellin X (**26**). In comparison with the other novel metabolite that was isolated from the same sample of soft coral (cladieunicellin W), the novel 6,7-secoeunicellin was found to display a much more potent inhibitory effect on the generation of superoxide anions and the release of elastase, with IC_50_ values of 7.18 and 7.83 μM, respectively [85].

#### 2.2.4. Xenicane-Type Diterpene

Xenicane diterpenes are characterized by a nine-membered ring often connected with a six-membered cyclic ether, resulting in a [7.4.0] bicyclic system [86] (Figure 7).

Tsitsixenicins A (**27**) and B (**28**), which were isolated from the South African soft coral *Capnella thyrsoidea*, represented the first xenicanes discovered from the family Neptheiidae. Both of these compounds inhibited the production of superoxide anion in human neutrophils, with 68% and 21% inhibitory rates at 1.25 µg/mL [87].

Asterolaurin D (**29**) is a new diterpene isolated from the soft coral *Asterospicularia laurae*, collected from southern Taiwan. The compound possesses a 2-oxabicyclo[7.4.0]tridecane ring system, which is a characteristic feature of xenicane skeletons. Asterolaurin D also contains a hemiacetal moiety at C-1, which might make it much more active than the other analogs in the same series of isolates and even the positive control genistein, with IC_50_ values for the inhibition of superoxide anion generation and elastase release of 23.6 and 18.7 μM, respectively [88].

#### 2.2.5. Miscellaneous Diterpenes

Echinohalimane A (**30**), a new diterpenoid, was isolated from *Echinomuricea* sp. It was the first halimane analog discovered from the phylum Cnidaria. The results of the in vitro experiment revealed that the new halimane-type diterpene exhibited potent anti-inflammatory activity, with an IC_50_ value of 0.38 μg/mL in inhibiting elastase release [89].

Cespitulin G (**31**), a new verticillane-type diterpene, was obtained in a chemical investigation of *Cespitularia taeniata*. It exhibited significant inhibitory activity against elastase release, with an IC_50_ value of 2.7 μg/mL, and against superoxide anion, with an IC_50_ value of 6.2 μg/mL [90].

Tortuosene A (**32**), which was isolated from the Formosan soft coral *Sarcophyton tortuosum*, possesses a new C-2/C-20-cyclized cembranoid skeleton. It showed a potent inhibitory effect against superoxide anion generation by human neutrophils with an IC_50_ value of 7.3 μM [75].

Two new norcembranoids, sinulerectols A (**33**) and B (**34**), were isolated from an extract of the marine soft coral *Sinularia erecta*. Both compounds were found to be potent agents in in vitro anti-inflammatory tests. Compound **33** showed IC_50_ values of 2.3 and 0.9 μM in the inhibition of superoxide generation and elastase release, respectively. In the same assays, the IC_50_ values of compound **34** were 8.5 and 3.8 μM [77].

A new capsosane, 7-epi-pavidolide D (**35**), was isolated from the marine soft coral *Klyxum flaccidum*, collected off the coast of Pratas island. The compound showed good activity against the release of elastase at 10 μM, with an inhibitory rate of 29.96% [78].

A new lobane, lobovarol G (**36**), along with a known analog, loba-8,10,13(15)-trien-14,17,18-triol-14,17-diacetate (**37**), was isolated from the ethyl acetate extract of the soft coral *Lobophytum varium*. Both compounds were found to be effective in the inhibition of elastase release with IC_50_ values of 18.8 and 6.9 μM. Lobovarol I (**38**), a new prenyleudesmane-type diterpene isolated from the same sample of the soft coral, inhibited the release of elastase with an IC_50_ value of 20 μM. The other known eudesmane derivative (**39**) was found to be active with IC_50_ values of 13.7 μM and 4.4 μM in superoxide anion generation and elastase release assays, respectively [91] (Figure 8).

### 2.3. Biscembranes

Biscembranes are marine natural products that are isolated from the soft coral of the genus *Sarcophyton*. They are characterized by a 14/6/14-membered tricyclic system [79] (Figure 9).

Glaucumolides A (**40**) and B (**41**), novel biscembranes composed of an unprecedented α,β-unsaturated ε-lactone, were isolated from the cultured soft coral *Sarcophyton glaucum*. Both compounds inhibited superoxide anion generation and elastase release with IC_50_ values of 2.79 and 3.97 μM, respectively [76].

Several biscembrane metabolites, which were isolated from the cultured soft coral *Sarcophyton trocheliophorum*, displayed significant anti-inflammatory effects in the in vitro assays. At a concentration of 10 μM, bistrochelides A (**42**) and B (**43**) were found to be good inhibitors of both superoxide anion generation (56.19% and 45.39%, respectively) and elastase release (48.61% and 38.67%, respectively) assays. Meanwhile, methyl tortuoate D (**44**) and ximaolide A (**45**) were less potent than the former two analogs at the same concentration, with 25.67% and 26.64% inhibitory activities against elastase release, respectively [79].

### 2.4. Steroids

Steroids are tetracyclic compounds that comprise three cyclohexane and cyclopentane rings fused, creating a distinctive perhydro-1,2-cyclopentenophenanthrene core skeleton. The introduction of different side chains and functional moieties, along with some modifications in the core skeleton generated various types of steroids. Methyl groups are normally found at C-10 and C-13, and an alkyl side chain may be also attached to C-17. The integrity of the steroidal scaffold can be disturbed by bond fissions, ring expansions, contractions, or removal of certain functionalities [92].

As per the investigation of related studies, sterols, 9,11-secosterols, gorgostane-type steroids, withanolide-type steroids, steroid glycosides, and several unclassified steroids have been recorded as potential neutrophilic-targeting anti-inflammatory steroids derived from octocorals.

#### 2.4.1. Sterols

Sterols represent a large group of natural steroids that bear a characteristic of a hydroxy group at C-3 in ring A of the steroid skeleton. Oxidation may occur in the side chain or the ring nucleus [55] (Figure 10). New steroids isolated from the Formosan soft coral *Klyxum flaccidum, k*lyflaccisteroids J (**46**) and M (**47**), displayed potent anti-inflammatory effects in activated human neutrophils. *K*lyflaccisteroid J significantly inhibited superoxide anion generation (an IC_50_ value of 5.64 μM) and elastase release (an IC_50_ value of 4.40 μM). Klyflaccisteroid M was inactive against the production of superoxide anions but was a potent inhibitor of elastase release, with an IC_50_ value of 5.84 μM [93,94].

A known steroid, which was identified as 5,6-epoxylitosterol (**48**), was obtained from the octocoral *Litophyton columnaris*. The compound significantly inhibited the generation of superoxide anion generation and the release of elastase, with IC_50_ values of 4.60 μM and 3.90 μM, respectively [95].

Three new polyoxygenated steroids, michosterols A–C (**49**–**51**), were isolated from the ethyl acetate extract of the soft coral *Lobophytum michaelae*. Michosterol A exhibited IC_50_ values of 7.1 and 4.5 μM against the superoxide anion production and elastase release, respectively. Michosterol C was considered a promising inhibitor of elastase release, with an IC_50_ value of only 0.9 μM. Michosterol B (**58**), which possesses a hydroperoxyl group at C-16 and an uncommon double bond between C-17 and C-20, showed the weakest effects in both in vitro assays [96].

#### 2.4.2. 9,11-Secosterols

9,11-secosterols (Figure 11) are a subtype of sterols, which are usually found in marine invertebrates, such as sponges, soft corals, and ascidians. Their structures are featured by the bond cleavage between C-9 and C-11 of ring C. The majority of 9,11-secosterols have a keto group attached to C-9 and a hydroxy group present at C-11 [97].

A known steroid, 5β,6β-epoxy-3β,11-dihydroxy-24-methylene-9,11-secocholestan-9-one (**52**), was obtained from the soft coral *Sinularia nanolobata*. It showed potent activity in in vitro anti-inflammatory assays. The IC_50_ values of the compound were 6.6 μM and 2.9 μM for the inhibition of superoxide anion generation and elastase release, respectively [98].

A series of unprecedented steroid skeletons were discovered from a gorgonian coral *Pinnigorgia* sp., which were assigned as pinnigorgiols A–E (**53**–**57**). The compounds contained a rare tricyclo[5,2,1,1]decane ring in their structures. Pinnigorgiols D and E were found to be the 11-*O*-acetyl derivative of pinnigorgiols A and B, respectively. The results of the in vitro experiment demonstrated that all the newly identified metabolites possessed anti-inflammatory potential. The IC_50_ values of pinnigorgiols A–E in the superoxide anion production assay were 4.0, 2.5, 2.7, 3.5, and 3.9 μM, respectively. Their inhibitory effects on the release of elastase were evidenced by IC_50_ values of 5.3, 3.1, 2.7, 2.1, and 1.6 μM [99,100].

Ten new 9,11-secosterols were identified in a recent study on the soft coral *Pinnigorgia* sp. More than half of the isolated steroids demonstrated impressive effects on neutrophilic inflammation, including pinnisterols A (**58**), C (**59**), E (**60**), F (**61**), H (**62**), and J (**63**). Except for pinnisterols E and F, which selectively inhibited elastase release and superoxide anion generation, with IC_50_ values of 2.33 and 5.52 μM, respectively, the remaining isolates displayed their effect in both anti-inflammatory assays. For superoxide anion production, their inhibitory effect was demonstrated by low IC_50_ values ranging from 2.33 to 3.89 μM. As inhibitors of elastase release, compounds **58**, **59**, **62**, and **63** exhibited IC_50_ values of 3.32, 2.81, 3.26, and 3.71 μM, respectively [101,102].

A detailed chemical investigation of the gorgonian coral *Pinnigorgia* sp. resulted in the isolation of a new bioactive sterol, 5α,6α-epoxy-(22*E*,24*R*)-3β,11-dihydroxy-9,11-secoergosta-7-en-9-one (**64**). This compound displayed inhibitory effects on the generation of superoxide anions and the release of elastase by human neutrophils, with IC_50_ values of 8.65 and 5.86 μM, respectively [103].

The 9,11-secosteroid targeting isolation conducted on the octocoral *Sinularia leptoclados* afforded six bioactive compounds, including two novel steroids, sinleptosterols A (**65**) and B (**66**), along with four known analogs, 8αH-3β,11-dihydroxy-24-methylene-9,11-secocholest-5-en-9-one (**67**), 8βH-3β,11-dihydroxy-24-methylene-9,11-secocholest-5-en-9-one (**68**), leptosterol A (**69**), and (24S)-3β,11-dihydroxy-24-methyl-9,11-secocholest-5-en-9-one (**70**). They exhibited potent anti-inflammatory effects in the in vitro assays. 8αH-3β,11-Dihydroxy-24-methylene-9,11-secocholest-5-en-9-one and 8βH-3β,11-dihydroxy-24-methylene-9,11-secocholest-5-en-9-one were the most potent agents in both superoxide anion production (IC_50_ values of 1.97 and 2.96 µM, respectively) and elastase release (IC_50_ values of 3.12 and 1.63 µM, respectively) assays. For the other four analogs, compound **70** was more active in the inhibition of superoxide anion generation than the others (IC_50_ value of 4.09 µM), but it was the weakest inhibitor of elastase release. Sinleptosterol A was less potent than sinleptosterol B in both assays. Their IC50 values were determined to be 7.07 and 4.68 µM in terms of superoxide anion inhibition, and 7.57 and 4.29 µM for their ability to inhibit elastase release, respectively. The known steroid leptosterol A was also considered a promising anti-inflammatory lead, as it significantly inhibited superoxide anion generation and elastase release, with IC_50_ values of 8.07 and 4.73 µM, respectively [104].

#### 2.4.3. Gorgostane-Type Steroids

Gorgostane steroids (Figure 12) are marine-derived sterols that possess a C30 skeleton with a characteristic three-membered ring present in the side chain attached to C-17 of the cyclopentane ring. In some cases, ring C of the steroidal skeleton loses its integrity due to the oxidative cleavage of the C-9/11 bond, resulting in derivatives named 9,11-secogorgosterols. These metabolites are characterized by the presence of a keto group at C-9 and a carboxyl or a hydroxyl moiety at C-11 [61].

The ethyl acetate extract of the soft coral *Klyxum flaccidum* yielded a series of new and known steroids. Some of these compounds were found to possess anti-inflammatory activity, including gorgost-5-ene-3β,9α,11α-triol (**71**), klyflaccisteroids C (**72**), D (**73**), F (**74**), K (**75**), and 3β,11-dihydroxy-9,11-secogorgost-5-en-9-one (**76**). Gorgost-5-ene-3β,9α,11α-triol was the least active metabolite, with only 27.7% inhibition against elastase release at 10 μM. The new gorgosteroid, klyflaccisteroid C, displayed IC_50_ values of 4.74 μM and 3.97 μM in the inhibition of superoxide anion production and elastase release, respectively. Klyflaccisteroid D, the C-7 oxidized product of klyflaccisteroid C, was less active in both assays when compared to its precursor, with a 30.9% inhibitory rate and an IC_50_ value of 5.37 μM. Klyflaccisteroid F was the first 9,11-secogorgosteroid 11-carboxylic acid isolated from natural sources, which was extremely potent in inhibiting superoxide anion generation (IC_50_ = 0.34 μM) and elastase release (IC_50_ = 0.35 μM). *K*lyflaccisteroid K, a new steroid possessing a 5,8-epidioxy-9-ene functional group, displayed a similar effect on the suppression of superoxide anion production but was more potent than *k*lyflaccisteroid J in inhibiting elastase release, with IC_50_ values of 5.83 μM and 1.55 μM, respectively. The IC_50_ values of the known secogorgosterol 3β,11-dihydroxy-9,11-secogorgost-5-en-9-one were determined to be 3.84 μM and 2.21 μM in the inhibition of superoxide anion production and elastase release, respectively [94,105].

#### 2.4.4. Withanolide-Type Steroids

Withanolides are a group of C28 polyoxygenated steroidal skeletons that bear a C-22/26 δ-lactone and C-26 or a C-23/26 γ-lactone in the side chain, which is attached to C-17 [106]. A series of withanolide steroids was isolated from the soft coral *Sinularia brassica*, including sinubrasolides A (**77**), H (**78**), J (**79**), K (**80**), and L (**81**) (Figure 13). The known metabolite sinubrasolide A was the most potent agent in the series, with IC_50_ values of 3.5 μM and 1.4 μM for superoxide anion generation and elastase release assays, respectively. Sinubrasolides H and K were found to be novel withanolides with a 16,23-oxo-bridged tetrahydropyran, and sinubrasolide K was the configurational isomer of sinubrasolide H. Regarding biological activity, sinubrasolide H was an effective inhibitor of elastase release (32.4% inhibition), while sinubrasolides J and K were active in the inhibition of superoxide anion generation with inhibitory rates of 32.1% and 34.3% at 10 μM, respectively. The remaining new withanolide, sinubrasolide L, was active in both assays, with inhibitory rates of 26.3% and 25.0% at 10 μM [106].

#### 2.4.5. Steroid Glycosides

A new bioactive metabolite, carijoside A (**82**), was isolated from an octocoral identified as *Carijoa* sp. The sterol glycoside displayed significant inhibitory effects on superoxide anion generation (IC_50_ = 1.8 μg/mL) and elastase release (IC_50_ = 6.8 μg/mL) by human neutrophils [107].

A new pragnane glycoside, hirsutosteroside A (**83**), was isolated from the soft coral *Cladiella hirsuta*. The compound was evaluated for its anti-inflammatory potential via in vitro tests. It was inactive in the inhibition of superoxide anion production but effectively suppressed the release of elastase with an IC_50_ value of 4.1 μM. At the same time, a known steroid, 24-methylenecholest-5-ene-3β,16β-diol-3-O-α-L-fucoside (**84**), was obtained from another soft coral, *Sinularia nanolobata*. The IC_50_ values of compound **84** were 18.6 μM and 10.1 μM in the inhibition of superoxide anion generation and elastase release assays, respectively [98].

*S*inubrasone A (**85**) is a novel steroid collected from the cultured soft coral *Sinularia brassica*. The compound possesses a methyl ester group attached to C-25 and a β-D-xylopyranose connected with C-22 via an O-glycoside bond. *S*inubrasone A significantly suppressed superoxide anion generation and elastase release (24.8% and 35.6% inhibition, respectively) at 10 μM [108] (Figure 14).

#### 2.4.6. Miscellaneous Steroids

6-*epi*-Yonarasterol B (**86**) is a new sterol that was the first steroid isolated from the gorgonian coral of the genus *Echinomuricea*. The compound showed a significant inhibitory effect on the generation of superoxide anions and the release of elastase by human neutrophils, with IC_50_ values of 2.98 and 1.13 μM, respectively [109].

A chemical investigation was conducted on the soft coral *Umbellulifera petasites*, for the first time, resulting in the isolation of three bioactive steroids with potent anti-inflammatory activity. Among these isolates, two new steroids, petasitosterones B (**87**) and C (**88**), demonstrated effective inhibitory activity of superoxide anion generation. The known steroid 5α-pregna-20-en-3-one (**89**) selectively suppressed the release of elastase. Their IC_50_ values were 4.43, 2.76, and 6.8 μM [110].

The Formosan soft coral *Klyxum flaccidum* yielded klyflaccisteroid L (**90**) that displayed notable effects in activated human neutrophils. Klyflaccisteroid L possessed an unusual 11-norsteroid skeleton and was the first representative of an 11-oxasteroid discovered in nature. The compound was inactive against superoxide anion production but demonstrated 25.17% inhibition in elastase release assay at 10 μM [94].

*S*inubrasones B–D (**91**–**93**), which are considered novel steroids with methyl ester groups, were collected from the cultured soft coral *Sinularia brassica*. *S*inubrasone B, possessing a tetrahydrofuran ring derived from the C-16/22 ether linkage, significantly suppressed superoxide anion generation and elastase release (19.4% and 39.0%, respectively) at 10 μM. *S*inubrasones C and D were more effective inhibitors of elastase release with approximately similar IC_50_ values of 6.6 μM and 6.5 μM, respectively. Sinubrasone D also exhibited a potent inhibitory effect against the generation of superoxide anion, with an IC_50_ value of 8.4 μM [108] (Figure 15).

### 2.5. Miscellaneous

5-(6-Hydroxy-2,5,7,8-tetramethyl-chroman-2-yl)-2-methyl-pentanoic acid methyl ester (**94**), an α-tocopherol derivative, was isolated from the soft coral *Sinularia arborea* for the first time. The bioassay revealed that the metabolite displayed a significant inhibitory effect on the generation of superoxide anion by human neutrophils with an IC_50_ value of 7.42 μM but was inactive toward elastase release [111].

Another new α-tocopherol derivative, hirsutocospiro A (**95**), was discovered in the soft coral *Cladiella hirsuta* with promising anti-inflammatory activity. The compound could be a promising candidate for the development of anti-inflammatory agents, as shown by the IC_50_ values of 4.1 μM and 3.7 μM for the suppression of superoxide anion production and elastase release, respectively [112].

(*Z*)-N-[2-(4-Hydroxyphenyl)ethyl]-3-methyldodec-2-enamide (**96**), a known nitrogen-containing compound isolated from the soft coral *Sinularia erecta*, was found to be a potent inhibitor of elastase release, with an IC_50_ value of 1.0 μM. The isolate also inhibited superoxide anion generation with an inhibitory rate of 48% at 10 μM [77].

Apo-9’-fucoxanthinone (**97**), isolated from a gorgonian coral *Pinnigorgia* sp., displayed a significant inhibitory effect on the release of elastase by human neutrophils, with an IC_50_ value of 5.75 μM [113] (Figure 16).

## 3. Preliminary Structure–Activity Relationship of the Octocoral-Derived Secondary Metabolites

As mentioned in the introduction, only 97 of 299 derivatives isolated from octocorals possess significant effects in the in vitro anti-inflammatory tests. The difference regarding their activities may be due to variations in their chemical structures. In this section, some examples regarding the impact of chemical variations on the difference in anti-inflammatory properties of octocoral-origin derivatives are introduced so as to provide the preliminary structure–activity relationship of these marine secondary metabolites.

In a chemical investigation of *Rumphella antipathies*, clovan-2,9-dione (**1**) was found to be the most potent compound among the isolates possessing a similar skeleton. This implies that the presence of the ketone group at C-2 plays a crucial role in the increased effect of sesquiterpenes. In the case of antipacids A and B, the difference is that the side chain at C-8 in the structure of antipacid B (**2**) is shorter than that of antipacid A, which may contribute to the enhanced anti-inflammatory potential of antipacid B in comparison with antipacid A. In addition, antipacid A and clovane-2β,9α-diol were inactive in the in vitro tests, whereas their combination via esterification resulted in rumphellolide L (**3**), which is a potent anti-inflammatory agent [65,66].

Among the polyoxygenated briaranes isolated from the octocoral *Briareum excavatum*, briarenol C was found to be inactive in the anti-inflammatory activity tests. In comparison to briarenols D (**9**) and E, the compound possesses a twisted boat conformation in the methylenencyclohexane ring, indicating that this configuration could significantly affect the anti-inflammatory property of briaranes [71].

Klymollin M (**20**) is one of the new diterpenes obtained from the organic extract of *Klyxum molle*. It was the first eunicelline bearing a phenylacetate group at C-6 and also exhibited the strongest inhibitory activity in the in vitro tests, as compared to other analogs in the study. This indicates that the presence of the substituent at C-6 may enhance the effects of klymollin M [81].

Cladieunicellin X (**26**) isolated from *Cladiella* sp. was found to demonstrate pronounced effects on superoxide anion generation and elastase release in activated neutrophils in comparison with cladieunicellin W. The difference between them is the presence of the methoxy group at C-6 in the structure of cladieunicellin X, implying that the functional group plays a critical role in the activity of these novel 6,7-secoeunicellins [85].

In comparison with the other xenicanes isolated from the same sample of the soft coral *Asterospicularia laurae*, the presence of the hemiacetal group in the chemical structure of asterolaurin D (**29**) was considered to be a critical factor in the enhancement in the inhibitory effects of the compound on elastase release in vitro [88].

Some modifications in the biscembrane skeletons of the derivatives originated from the cultured soft coral Sarcophyton trocheliophorum and led to the different extents of their inhibitory effects on superoxide anion generation and elastase release. A 6,7-dihydrooxepin-2(5H)-one moiety and a saturated γ-lactone ring are the common features of the four most potent compounds, glaucumolides A (**40**) and B (**41**) and bistrochelides A (**42**) and B (**43**). In addition to the characteristic functional groups, the bioactivities of the four compounds also varied when compared to each other due to some minor variations in their configurations. Particularly, glaucumolide B exhibits 11Z and 22E double bonds, differing from the E configuration observed at Δ^11(12)^ and Δ^22(23)^ in glaucumolide A, as well as the Z configuration at Δ^11(12)^ and Δ^22(23)^ in bistrochelide A. Furthermore, a comparison with glaucumolide B reveals that glaucumolide B features a substitution of the 11,12 double bond with a 10,11 double bond. This alteration implies that the slight modification in bistrochelide B, leading to the replacement of the 11,12-double bond, could be accountable for the diminished anti-inflammatory effect observed in it [79].

The two sterols, 5,6-epoxylitosterol (**48**) and litosterol, were obtained from the same sample of *Nephthea columnaris*. However, litosterol was inactive in the anti-inflammatory tests, whereas 5,6-epoxylitosterol showed highly potent effects on superoxide anion generation and elastase release. The difference in their properties implies the crucial role of the 5β,6β-epoxy group in increasing the activity of 5,6-epoxylitosterol [95].

Pinnigorgiols A–E (**53**–**57**) isolated from *Pinnigorgia* sp. demonstrated promising anti-inflammatory activities. Nonetheless, pinnigorgiol E (**57**) was the most effective inhibitor of elastase release among the isolates, which suggests the important role of an acetoxy substituent at C-11 and the absence of the C-22/23 double bond in the structure of this compound novel compound 9,11-secosterol [99,100].

The gorgonian coral *Pinnigorgia* sp. yielded a potent anti-inflammatory agent and an inactive analog. The difference in their pharmacological properties was believed to be driven by the acetoxy group at C-3 in ring A of pinnisterol C (**59**), as compared to the structure of pinnisterol B [101].

Among the metabolites isolated from the octocoral *Sinularia leptoclados*, 3β,11-dihydroxy-9,11-secogorgost-5-en-9-one presented activity at a concentration greater than 10 μM in both anti-inflammatory in vitro tests. By comparing the chemical structures of these analogs, it was found that 3β,11-dihydroxy-9,11-secogorgost-5-en-9-one possessed a unique gorgosterol side chain, which perhaps accounted for its nullified activities [104].

In a study on the chemical constituents of *Umbellulifera petasites*, petasitosterone C (**88**) was the steroid that possessed a rare A/B spiro[4,5]decane ring system and displayed the strongest activities against superoxide anion generation, as compared to the other steroids isolated from the same sample. This indicates that the unique chemical feature of the compound might be related to its enhanced activity [110].

**Table 1 marinedrugs-21-00456-t001:** Potent anti-inflammatory compounds derived from octocorals.

Compound Name	Novelty	Chemical Classification	Source	Inhibitory Effects		Ref.
				O^•−^ Generation	Elastase Release	
Clovan-2,9-dione (1)	New	Sesquiterpene	*Rumphella antipathies*	IC_50_ = 2.72 ± 0.93 μg/mL	IC_50_ = 6.73 ± 0.85 μg/mL	[65]
Antipacid B (2)	Novel	Sesquiterpene	*Rumphella antipathies*	IC_50_ = 11.22 μM	IC_50_ = 23.53 μM	[66]
Rumphellolide L (3)	New	Sesquiterpene	*Rumphella antipathies*	Inh% = 19.57 ± 3.69 (10 μg/mL)	IC_50_ = 7.63 μM	[66]
Junceol A (4)	New	Briarane-type diterpene	*Junceella juncea*	Inh% = 45.64 % (10 μg/mL)		[68]
Junceol B (5)	New	Briarane-type diterpene	*Junceella juncea*	Inh% = 159.60 % (10 μg/mL)		[68]
Junceol C (6)	New	Briarane-type diterpene	*Junceella juncea*	Inh% = 124.14 % (10 μg/mL) [68]		[68]
Briarenolide F (7)	New	Briarane-type diterpene	*Briareum* sp.	IC_50_ = 3.82 ± 0.45 μg/mL	Inh% = 27.48 ± 6.60 (10 μg/mL)	[69]
Briarenolide J (8)	Novel	Briarane-type diterpene	*Briareum* sp.	IC_50_ = 14.98 μM	IC_50_ = 9.96 μM	[70]
Briarenol D (9)	New	Briarane-type diterpene	*Briareum excavatum*		IC_50_ = 4.65 μM	[71]
Juncin Z (10)	New	Briarane-type diterpene	*Junceella fragilis*	Inh% = 25.56% (10 μM)		[72]
Lobocrassin B (11)	New	Cembrane-type diterpene	*Lobophytum crassum*	IC_50_ = 4.8 ± 0.7 μg/mL	IC_50_ = 4.9 ± 0.4 μg/mL	[73]
Arbolide C (12)	New	Cembrane-type diterpene	*Sinularia arborea*		IC_50_ = 5.13 μg/mL	[74]
Emblide (13)	Known	Cembrane-type diterpene	*Sarcophyton tortuosum*		Inh% = 29.2 ± 6.1 (10 μM)	[75]
Isosarcophytonolide D (14)	Known	Cembrane-type diterpene	*Sarcophyton glaucum*	Inh% = 12.40 ± 2.56 (10 μM)	Inh% = 27.12 ± 3.08 (10 μM)	[76]
Sinulerectol C (15)	New	Cembrane-type diterpene	*Sinularia erecta*	Inh% = 24 ± 7 (10 μM)	Inh% = 33 ± 3 (10 μM)	[77]
14-Deoxycrassin (16)	Known	Cembrane-type diterpene	*Sinularia flexibilis*	IC_50_ = 10.8 ± 0.38 μM	IC_50_ = 11.0 ± 1.52 μM	[59]
Flaccidodioxide (17)	New	Cembrane-type diterpene	*Klyxum flaccidum*	Inh% = 8.88 ± 3.33 (10 μM)	Inh% = 27.18 ± 4.05 (10 μM)	[78]
14-O-Acetylsarcophytol B (18)	Known	Cembrane-type diterpene	*Klyxum flaccidum*	Inh% = 11.95 ± 2.53 (10 μM)	IC_50_ = 7.22 ± 0.85 μM	[78]
Cladielloide B (19)	New	Eunicellin-type diterpene	*Cladiella* sp.	IC_50_ = 5.9 ± 0.7 μg/mL	IC_50_ = 6.5 ± 1.9 μg/mL	[80]
Klymollin M (20)	New	Eunicellin-type diterpene	*Klyxum molle*	IC_50_ = 3.13 ± 0.39 μM	IC_50_ = 2.92 ± 0.27 μM	[81]
Krempfielin K (21)	New	Eunicellin-type diterpene	*Cladiella krempfi*		Inh% = 45.51 ± 2.69 (10 μM)	[82]
Krempfielin M (22)	New	Eunicellin-type diterpene	*Cladiella krempfi*		Inh% = 27.30 ± 5.42 (10 μM)	[82]
Krempfielin N (23)	New	Eunicellin-type diterpene	*Cladiella krempfi*		IC_50_ = 4.94 ± 1.68 μM	[83]
Krempfielin P (24)	New	Eunicellin-type diterpene	*Cladiella krempfi*	Inh% = 23.32% ± 5.88 (10 μM)	Inh% = 35.54 ± 3.17 (10 μM)	[83]
Sclerophytin B (25)	Known	Eunicellin-type diterpene	*Cladiella* sp.	Inh% = 28.12 ± 3.61 (10 μM)	Inh% = 16.37 ± 8.14 (10 μM)	[84]
Cladieunicellin X (26)	Novel	Eunicellin-type diterpene	*Cladiella* sp.	IC_50_ = 7.18 ± 1.20 μM	IC_50_ = 7.83 ± 0.83 μM	[85]
Tsitsixenicin A (27)	New	Xenicane-type diterpene	*Capnella thyrsoidea*	Inh% = 68% at 1.25 µg/mL		[87]
Tsitsixenicin B (28)	New	Xenicane-type diterpene	*Capnella thyrsoidea*	Inh% = 21% at 1.25 µg/mL		[87]
Asterolaurin D (29)	New	Xenicane-type diterpene	*Asterospicularia laurae*	IC_50_ = 23.6 μM	IC_50_ = 18.7 μM	[88]
Echinohalimane A (30)	New	Halimane-type diterpene	*Echinomuricea* sp.	Inh% = 20.55 ± 5.18 (10 μg/mL)	IC_50_ = 0.38 ± 0.14 μg/mL	[89]
Cespitulin G (31)	New	Verticillane-type diterpene	*Cespitularia taeniata*	IC_50_ = 6.2 μg/mL	IC_50_ =2.7 μg/mL	[90]
Tortuosene A (32)	New	Tortuosane-type diterpene	*Sarcophyton tortuosum*	IC_50_ = 7.3 ± 0.8 μM		[75]
Sinulerectol A (33)	New	Norcembrane-type diterpene	*Sinularia erecta*	IC_50_ = 2.3 ± 0.4 μM	IC_50_ = 0.9 ± 0.1 μM	[77]
Sinulerectol B (34)	New	Norcembrane-type diterpene	*Sinularia erecta*	IC_50_ = 8.5 ± 0.3 μM	IC_50_ = 3.8 ± 0.6 μM	[77]
7-Epi-Pavidolide D (35)	New	Capnosane-type diterpene	*Klyxum flaccidum*	Inh% = 24.46 ± 6.99 (10 μM)	Inh% = 29.96 ± 6.14 (10 μM)	[78]
Lobovarol G (36)	New	Lobane-type diterpene	*Lobophytum varium*	Inh% = 18.1 ± 4.0 (10 μg/mL)	IC_50_ = 18.8 ± 1.8 μM	[91]
Loba-8,10,13(15)-trien-14,17,18-triol-14,17-diacetate (37)	Known	Lobane-type diterpene	*Lobophytum varium*	Inh% = 46.5 ± 5.8 (10 μg/mL)	IC_50_ = 6.9 ± 2.7 μM	[91]
Lobovarol I (38)	New	Prenyleudesmane-type diterpene	*Lobophytum varium*	Inh% = 40.2 ± 7.3 (10 μg/mL)	IC_50_ = 20.0 ± 3.0 μM	[91]
An eudesmane derivative (39)	Known	Prenyleudesmane-type diterpene	*Lobophytum varium*	IC_50_ = 13.7 ± 4.4 μM	IC_50_ = 4.4 ± 0.7 μM	[91]
Glaucumolide A (40)	Novel	Biscembrane	*Sarcophyton glaucum*	IC_50_ = 2.79 ± 0.66 μM	IC_50_ = 3.97 ± 0.10 μM	[76]
Glaucumolide B (41)	Novel	Biscembrane	*Sarcophyton glaucum*	IC_50_ = 2.79 ± 0.32 μM	IC_50_ = 3.97 ± 0.10 μM	[76]
Bistrochelide A (42)	Known	Biscembrane	*Sarcophyton trocheliophorum*	IC_50_ = 8.29 ± 0.48 μM	Inh% = 48.61 ± 0.96 (10 μM)	[79]
Bistrochelide B (43)	Known	Biscembrane	*Sarcophyton trocheliophorum*	Inh% = 45.39 ± 4.30 (10 μM)	Inh% = 38.67 ± 4.81 (10 μM)	[79]
Methyl tortuoate D (44)	Known	Biscembrane	*Sarcophyton trocheliophorum*	Inh% = 17.61 ± 1.99 (10 μM)	Inh% = 25.67 ± 5.27 (10 μM)	[79]
Ximaolide A (45)	Known	Biscembrane	*Sarcophyton trocheliophorum*	Inh% = 19.69 ± 5.00	Inh% = 26.64 ± 5.02 (10 μM)	[79]
Klyflaccisteroid J (46)	New	Sterol	*Klyxum flaccidum*	IC_50_ = 5.64 ± 0.41 μM	IC_50_ = 4.40 ± 0.19 μM	[93]
Klyflaccisteroid M (47)	New	Sterol	*Klyxum flaccidum*	Inh% = 12.61 ± 1.70 (10 μM)	IC_50_ = 5.84 ± 0.33 10 μM	[94]
5,6-Epoxylitosterol (48)	Known	Sterol	*Litophyton columnaris*	IC_50_ = 4.60 ± 0.85 μM	IC_50_ = 3.90 ± 0.88 μM	[95]
Michosterol A (49)	New	Sterol	*Lobophytum michaelae*	IC_50_ = 7.1 ± 0.3 μM	IC_50_ = 4.5 ± 0.9 μM	[96]
Michosterol B (50)	New	Sterol	*Lobophytum michaelae*	Inh% = 14.7 ± 5.7 (10 μM)	Inh% = 31.8 ± 5.0 (10 μM)	[96]
Michosterol C (51)	New	Sterol	*Lobophytum michaelae*	Inh% = 17.8 ± 2.8 (10 μM)	IC_50_ = 0.9 ± 0.1 μM	[96]
5β,6β-Epoxy-3β,11-dihydroxy-24-methylene-9,11-secocholestan-9-one (52)	Known	Secosterol	*Sinularia nanolobata*	IC_50_ = 6.6 ± 0.6 μM	IC_50_ = 2.9 ± 0.5 μM	[98]
Pinnigorgiol A (53)	Novel	Secosterol	*Pinnigorgia* sp.	IC_50_ = 4.0 μM	IC_50_ = 5.3 μM	[99]
Pinnigorgiol B (54)	Novel	Secosterol	*Pinnigorgia* sp.	IC_50_ = 2.5 μM	IC_50_ = 3.1 μM	[99]
Pinnigorgiol C (55)	Novel	Secosterol	*Pinnigorgia* sp.	IC_50_ = 2.7 μM	IC_50_ = 2.7 μM	[99]
Pinnigorgiol D (56)	New	Secosterol	*Pinnigorgia* sp.	IC_50_ = 3.5 μM	IC_50_ = 2.1 μM	[100]
Pinnigorgiol E (57)	New	Secosterol	*Pinnigorgia* sp.	IC_50_ = 3.9 μM	IC_50_ = 1.6 μM	[100]
Pinnisterol A (58)	New	Secosterol	*Pinnigorgia* sp.	IC_50_ = 2.33 μM	IC_50_ = 3.32 μM	[101]
Pinnisterol C (59)	New	Secosterol	*Pinnigorgia* sp.	IC_50_ = 2.50 μM	IC_50_ = 2.81 μM	[101]
Pinnisterol E (60)	New	Secosterol	*Pinnigorgia* sp.		IC_50_ = 2.33 ± 0.27 μM	[102]
Pinnisterol F (61)	New	Secosterol	*Pinnigorgia* sp.	IC_50_ = 5.52 ± 1.06 μM		[102]
Pinnisterol H (62)	New	Secosterol	*Pinnigorgia* sp.	IC_50_ = 3.26 ± 0.33 μM	IC_50_ = 2.59 ± 0.29 μM	[102]
Pinnisterol J (63)	New	Secosterol	*Pinnigorgia* sp.	IC_50_ = 3.71 ± 0.51 μM	IC_50_ = 3.89 ± 1.16 μM	[102]
5α,6α-Epoxy-(22E,24R)-3β,11-dihydroxy-9,11-secoergosta-7-en-9-one (64)	New	Secosterol	*Pinnigorgia* sp.	IC_50_ = 8.65 ± 0.19 μM	IC_50_ = 5.86 ± 0.95 μM	[103]
Sinleptosterol A (65)	Novel	Secosterol	*Sinularia leptoclados*	IC_50_ = 7.07 ± 0.52 µM	IC_50_ = 7.57 ± 0.40 µM	[104]
Sinleptosterol B (66)	Novel	Secosterol	*Sinularia leptoclados*	IC_50_ = 4.68 ± 0.57 µM	IC_50_ = 4.29 ± 0.25 µM	[104]
8αH-3β,11-Dihydroxy-24-methylene-9,11-secocholest-5-en-9-one (67)	Known	Secosterol	*Sinularia leptoclados*	IC_50_ = 1.97 ± 0.12 µM	IC_50_ = 3.12 ± 0.07 µM	[104]
8βH-3β,11-dihydroxy-24-methylene-9,11-secocholest-5-en-9-one (68)	Known	Secosterol	*Sinularia leptoclados*	IC_50_ = 2.96 ± 0.91 µM	IC_50_ = 1.63 ± 0.15 µM	[104]
Leptosterol A (69)	Known	Secosterol	*Sinularia leptoclados*	IC_50_ = 8.07 ± 0.53 µM	IC_50_ = 4.73 ± 0.57 µM	[104]
(24S)-3β,11-Dihydroxy-24-methyl-9,11-secocholest-5-en-9-one (70)	Known	Secosterol	*Sinularia leptoclados*	IC_50_ = 4.09 ± 0.50 µM	Inh% = 25.38 ± 6.68 at 10 µM	[104]
Gorgost-5-ene-3β,9α,11α-triol (71)	Known	Gorgostane-type steroid	*Klyxum flaccidum*	Inh% = 10.52 ± 2.71 (10 μM)	Inh% = 27.70 ± 5.29 (10 μM)	[105]
Klyflaccisteroid C (72)	New	Gorgostane-type steroid	*Klyxum flaccidum*	IC_50_ = 4.78 ± 0.87 μM	IC_50_ = 3.97 ± 0.10 μM	[105]
Klyflaccisteroid D (73)	New	Gorgostane-type steroid	*Klyxum flaccidum*	Inh% = 30.9 ± 4.68 (10 μM)	IC_50_ = 5.37 ± 0.20 μM	[105]
Klyflaccisteroid F (74)	New	Gorgostane-type steroid	*Klyxum flaccidum*	IC_50_ = 0.34 ± 0.01 μM	IC_50_ = 0.35 ± 0.04 μM	[105]
Klyflaccisteroid K (75)	New	Gorgostane-type steroid	*Klyxum flaccidum*	IC_50_ = 5.83 ± 0.62 μM	IC_50_ = 1.55 ± 0.21 μM	[94]
3β,11-Dihydroxy-9,11-secogorgost-5-en-9-one (76)	Known	Gorgostane-type steroid	*Klyxum flaccidum*	IC_50_ = 3.84 ± 0.41 μM	IC_50_ = 2.21 ± 0.59 μM	[105]
Sinubrasolide A (77)	Known	Withanolide-type steroid	*Sinularia brassica*	IC_50_ = 3.5 ± 0.9 μM	IC_50_ = 1.4 ± 0.1 μM	[106]
Sinubrasolide H (78)	New	Withanolide-type steroid	*Sinularia brassica*	Inh% = 14.4 ± 3.1 (10 μM)	Inh% = 32.4 ± 5.6 (10 μM)	[106]
Sinubrasolide J (79)	New	Withanolide-type steroid	*Sinularia brassica*	Inh% = 32.1 ± 5.3 (10 μM)	Inh% = 9.5 ± 5.2 (10 μM)	[106]
Sinubrasolide K (80)	New	Withanolide-type steroid	*Sinularia brassica*	Inh% = 34.3 ± 6.6 (10 μM)	Inh% = 11.4 ± 3.2 (10 μM)	[106]
Sinubrasolide L (81)	New	Withanolide-type steroid	*Sinularia brassica*	Inh% = 26.3 ± 0.7 (10 μM)	Inh% = 25.0 ± 1.3 (10 μM)	[106]
Carijoside A (82)	New	Steroid glycoside	*Carijoa* sp.	IC_50_ = 1.8 μg/mL	IC_50_ = 6.8 μg/mL	[107]
Hirsutosteroside A (83)	New	Steroid glycoside	*Cladiella hirsuta*		IC_50_ = 4.1 ± 0.1 μM	[98]
24-Methylenecholest-5-ene-3β,16β-diol-3-O-α-L-fucoside (84)	Known	Steroid glycoside	*Sinularia nanolobata*	IC_50_ = 18.6 ± 1.5 μM	IC_50_ = 10.1 ± 0.8 μM	[98]
Sinubrasone A (85)	New	Steroid glycoside	*Sinularia brassica*	Inh% = 24.8 ± 6.5 (10 μM)	Inh% = 35.6 ± 1.3 (10 μM)	[108]
6-Epi-yonarasterol B (86)	New	Miscellaneous steroid	*Echinomuricea* sp.	IC_50_ = 2.98 ± 0.29 μg/mL	IC_50_ = 1.13 ± 0.55 μg/mL	[109]
Petasitosterone B (87)	New	Miscellaneous steroid	*Umbellulifera petasites*	IC_50_ = 4.43 ± 0.23 μM		[110]
Petasitosterone C (88)	New	Miscellaneous steroid	*Umbellulifera petasites*	IC_50_ = 2.76 ± 0.92 μM		[110]
5α-Pregna-20-en-3-one (89)	Known	Miscellaneous steroid	*Umbellulifera petasites*		IC_50_ = 6.80 ± 0.18 μM	[110]
Klyflaccisteroid L (90)	New	Miscellaneous steroid	*Klyxum flaccidum*		Inh% = 25.17 ± 6.73 (10 μM)	[94]
Sinubrasone B (91)	New	Miscellaneous steroid	*Sinularia brassica*	Inh% = 19.4 ± 5.0 (10 μM)	Inh% = 39.0 ± 2.3 (10 μM)	[108]
Sinubrasone C (92)	New	Miscellaneous steroid	*Sinularia brassica*	Inh% = 27.7 ± 1.3 (10 μM)	IC50 = 6.6 ± 1.7 μM	[108]
Sinubrasone D (93)	New	Miscellaneous steroid	*Sinularia brassica*	IC_50_ = 8.4 ± 1.1 μM	IC50 = 6.5 ± 1.1 μM	[108]
5-(6-Hydroxy-2,5,7,8-tetramethyl-chroman-2-yl)-2-methyl-pentanoic acid methyl ester (94)	New	α-tocopherol derivative	*Sinularia arborea*	IC_50_ = 7.42 μM		[111]
Hirsutocospiro A (95)	New	α-tocopherol derivative	*Cladiella hirsuta*	IC_50_ = 4.1 ± 1.1 μM	IC50 = 3.7 ± 0.3 μM	[112]
(Z)-N-[2-(4-Hydroxyphenyl)ethyl]-3-methyldodec-2-enamide (96)	Known	Nitrogen-containing compound	*Sinularia erecta*	Inh% = 48 ± 2 (10 μM)	IC50 = 1.0 ± 0.2 μM	[77]
Apo-9’-fucoxanthinone (97)	Known	Allenic norterpenoid ketone	*Pinnigorgia* sp.		IC_50_ = 5.75 μM	[113]

## 4. Conclusions and Perspectives

The generation of ROS is a crucial component of the antimicrobial effectiveness exhibited by neutrophils. Nevertheless, the excessive production or insufficient removal of ROS can lead to harm to cells and tissues; the oxidation and alteration of DNA, lipids, and proteins; the formation of autoimmune neutrophil extracellular traps (NETs); and the creation of autoantibodies. Indeed, ROS accumulation has been found to be related to psoriasis, a chronic systemic inflammation. The excessive generation of ROS activates dendritic cells so as to display antigens to T cells, leading to an imbalance between T helper (Th)1 and Th2 cells. The equilibrium disruption triggers the growth of keratinocytes and encourages angiogenesis. Subsequently, various inflammatory pathways, such as mitogen-activated protein kinase (MAPK), nuclear factor-kappa B (NF-κB), or those related to Janus kinase-signal transducer and activator of transcription proteins (JAK-STAT), are also activated by ROS [13]. These findings suggest that ROS generation of activated neutrophils could be a potential target for the suppression of inflammation. Therefore, respiratory burst inhibitors have emerged as promising solutions for the treatment of inflammation.

Elastase enzyme is a critical component of extracellular traps which involves in microbicidal activity of neutrophils. Its function links to the digestion of invading microorganisms [114]. However, in case of overproduction, neutrophil elastase can promote prolonged inflammation, which may lead to worsening diseases. Numerous studies have discovered different mechanisms related to chronic lung diseases of extracellular neutrophil elastase, including inducing airway mucus obstruction, modifying cellular differentiation and cellular fate, activating pro-inflammatory signaling, and impairing the innate immune system. The supporting evidence for the crucial role of neutrophil elastase in the pathogenesis of chronic lung diseases related to sustained inflammation, such as bronchopulmonary dysplasia, bronchiectasis, chronic obstructive pulmonary disease, and cystic fibrosis lung disease, has driven more studies regarding antineutrophil elastase agents [115].

Based on a thorough literature investigation, octocorals have proven to be a prolific source of anti-inflammatory natural products that can selectively target neutrophil activities, in particular, superoxide anion generation and elastase release. Up to the present, seven isolates have been discovered to exhibit potent activities in in vitro experiments involving superoxide anion generation and elastase release assays. Half the identified compounds were steroid derivatives, followed by diterpenes, which were the most abundant secondary metabolites isolated from these marine organisms. Several other compounds belonging to sesquiterpenes, biscembrane, and other chemical classes were also demonstrated as potential agents in terms of anti-inflammation. More than 65 out of 97 octocoral derivatives exhibited potent anti-inflammatory properties with IC_50_ values less than 10 μM or equivalent inhibitory rates. More than 50% of the potent compounds were steroids, especially secosterols.

Currently, there has been no in vivo study conducted on marine-derived compounds for the investigation of their mechanisms of action on neutrophils. Nonetheless, the aforementioned potentials of respiratory burst and elastase inhibitors in the resolution of inflammation-related diseases have demonstrated opportunities for the development of these marine natural products as effective therapeutic agents that target neutrophilic inflammation in the future. As a result, detailed studies on the mechanisms of action need to be conducted to support the subsequent development of novel neutrophil-targeting anti-inflammatory agents. The structure–activity relationships also need to be systematically investigated so as to develop the most potent derivatives with the least side effects.

In summary, the discovery and development of lead compounds targeting neutrophilic inflammation is an intriguing aspect of natural product research in recent decades. Octocorals represent a vast reservoir of promising secondary metabolites with interesting biological activity, especially anti-inflammatory effects. More research is needed to fully utilize these new compounds as safe and effective treatments.

## Figures and Tables

**Figure 1 marinedrugs-21-00456-f001:**
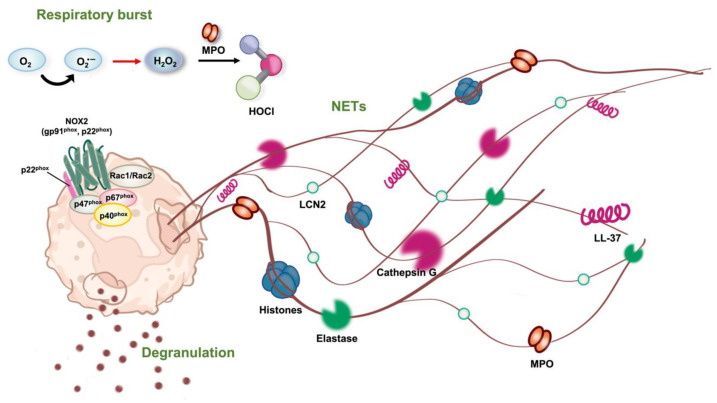
The inflammatory mechanisms employed by neutrophils. When neutrophils are activated, they utilize respiratory burst, degranulation, and the formation of neutrophil extracellular traps (NETs) as the primary mechanisms to elicit inflammation.

**Figure 2 marinedrugs-21-00456-f002:**
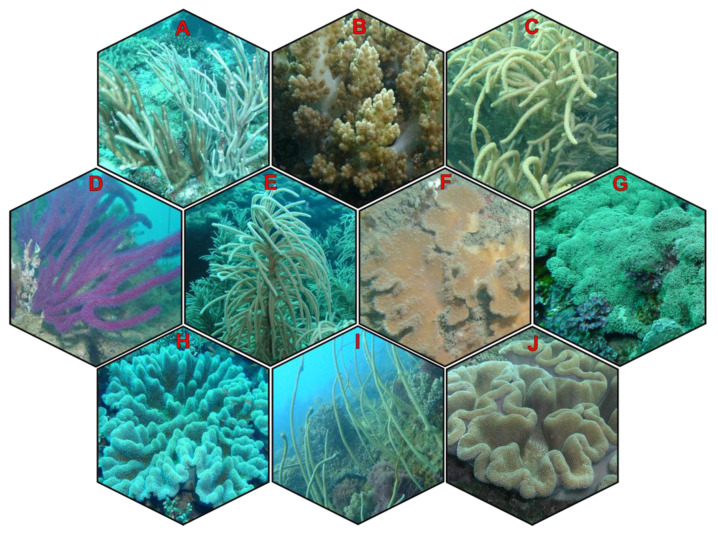
Soft corals that are reported to regulate neutrophilic inflammation. (**A**) *Rumphella* sp.; (**B**) *Litophyto*n sp.; (**C**) *Sinularia flexibilis*; (**D**) *Echinomuricea* sp.; (**E**) *Pinnigorgia* sp.; (**F**) *Sinularia brassica*; (**G**) *Briareum* sp.; (**H**) *Lobophytum* sp.; (**I**) *Junceella fragilis*; (**J**) *Sarcophyton* sp.

**Figure 3 marinedrugs-21-00456-f003:**
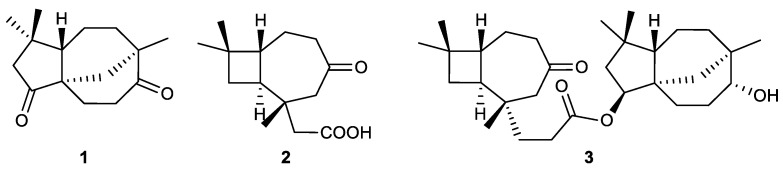
Sesquiterpenes (**1**–**3**) isolated from *Rumphella antipathies*.

**Figure 4 marinedrugs-21-00456-f004:**
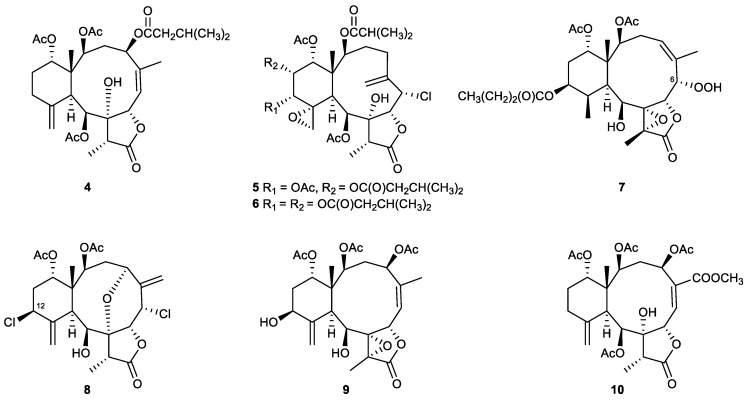
Briaranes (**4**–**10**) isolated from the octocorals *Junceella* sp. and *Briareum* sp.

**Figure 5 marinedrugs-21-00456-f005:**
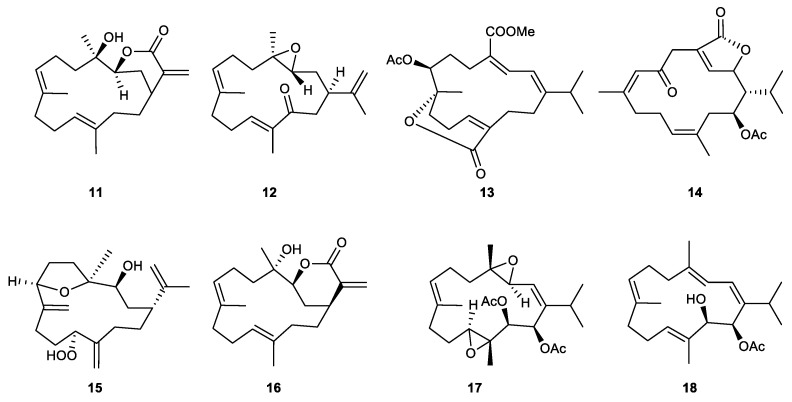
Cembranes (**11**–**18**) isolated from octocorals.

**Figure 6 marinedrugs-21-00456-f006:**
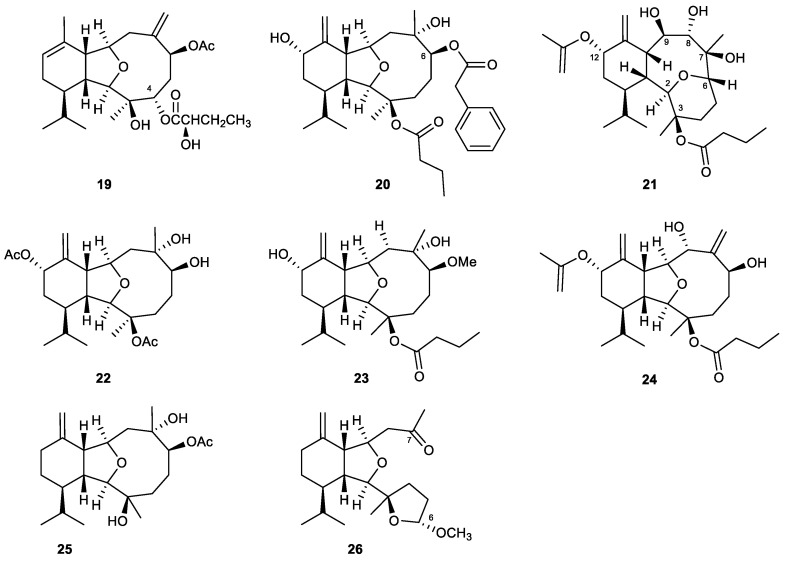
Eunicellines (**19**–**26**) isolated from *Cladiella* sp. and *Klyxum molle*.

**Figure 7 marinedrugs-21-00456-f007:**
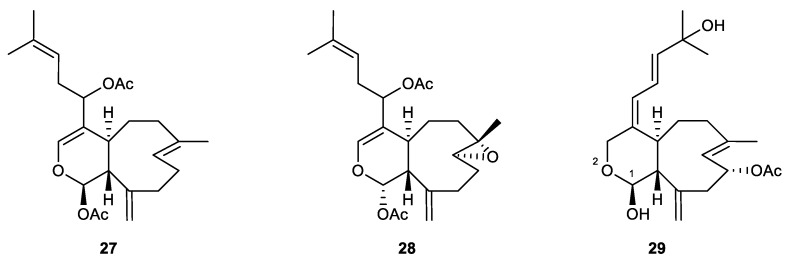
Xenicanes (**27**–**29**) isolated from soft corals.

**Figure 8 marinedrugs-21-00456-f008:**
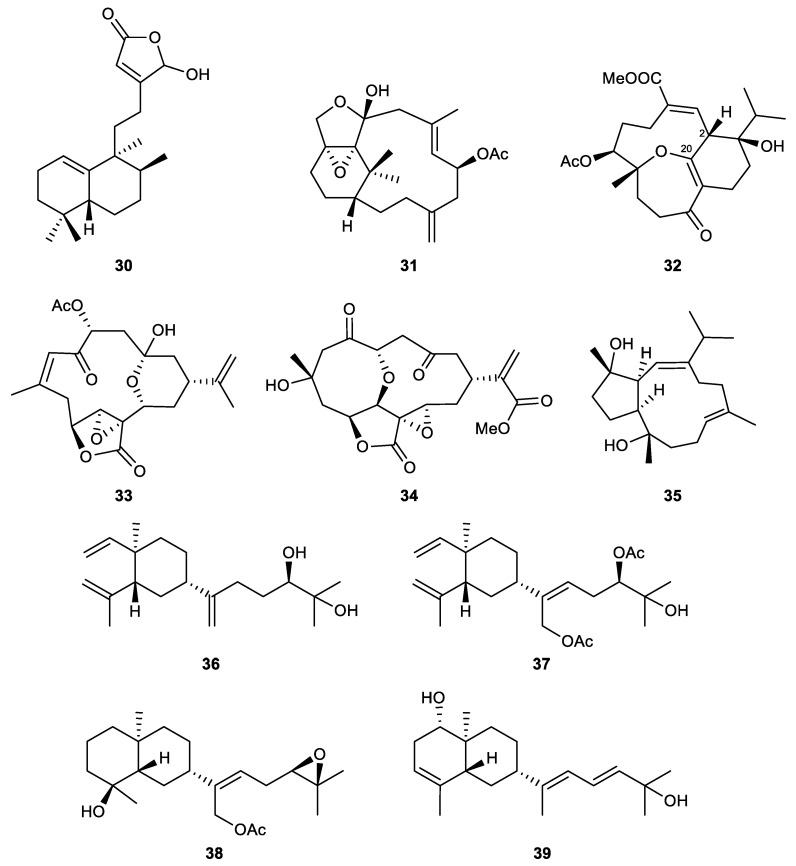
Anti-inflammatory diterpenoids (**30**–**39**) isolated from different soft corals.

**Figure 9 marinedrugs-21-00456-f009:**
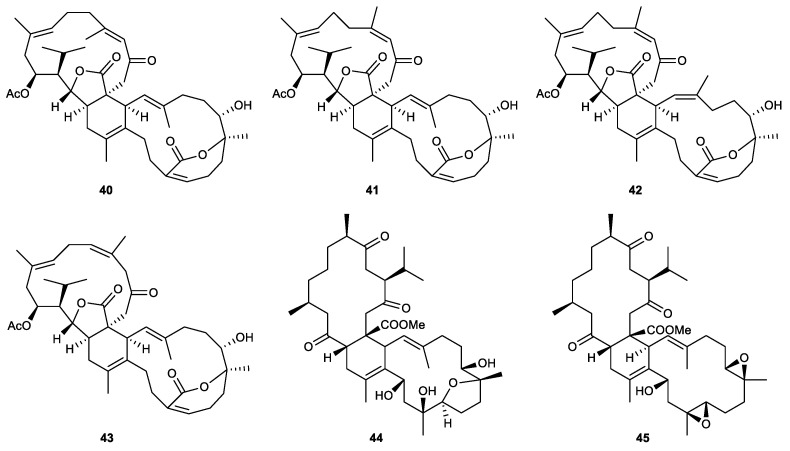
Biscembranes (**40**–**45**) isolated from the genus *Sarcophyton*.

**Figure 10 marinedrugs-21-00456-f010:**
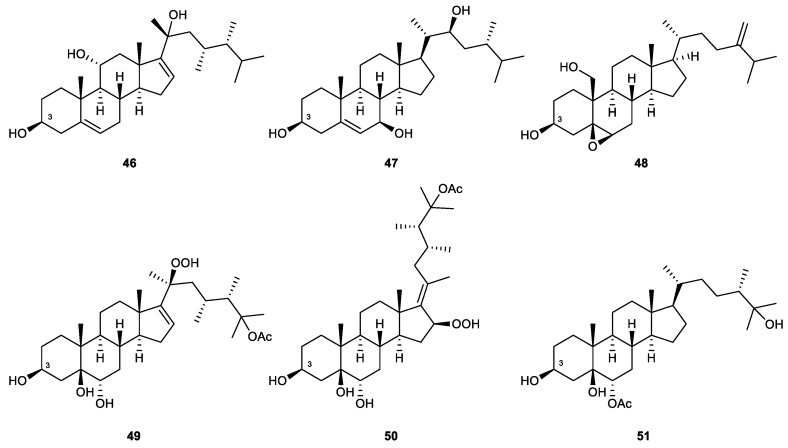
Sterols (**46**–**51**) isolated from octocorals.

**Figure 11 marinedrugs-21-00456-f011:**
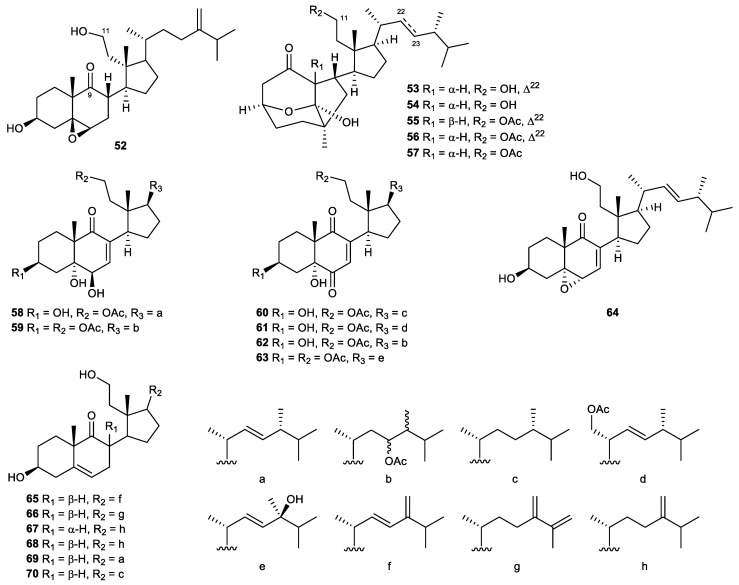
9,11-secosterols (**52**–**70**) isolated from the genera *Sinularia* and *Pinnigorgia*.

**Figure 12 marinedrugs-21-00456-f012:**
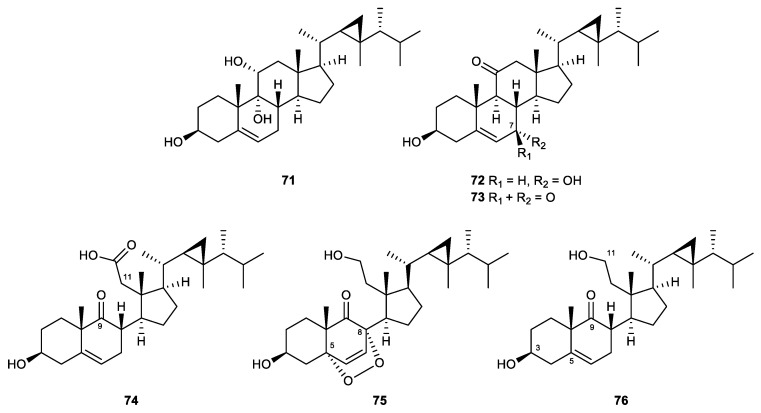
Gorgostane steroids (**71**–**76**) isolated from the soft coral *Klyxum flaccidum*.

**Figure 13 marinedrugs-21-00456-f013:**
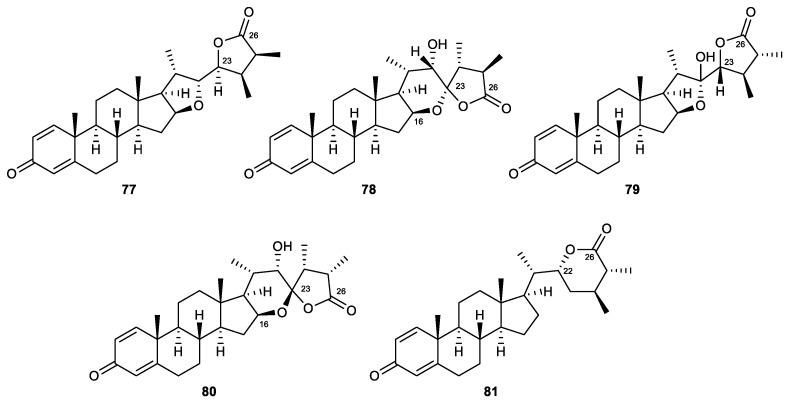
Withanolide steroids (**77**–**81**) isolated from the soft coral *Sinularia brassica*.

**Figure 14 marinedrugs-21-00456-f014:**
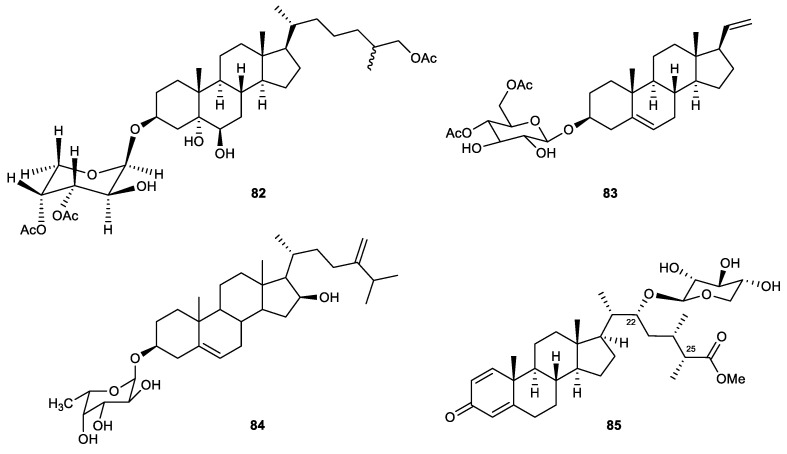
Steroid glycosides (**82**–**85**) isolated from the soft corals.

**Figure 15 marinedrugs-21-00456-f015:**
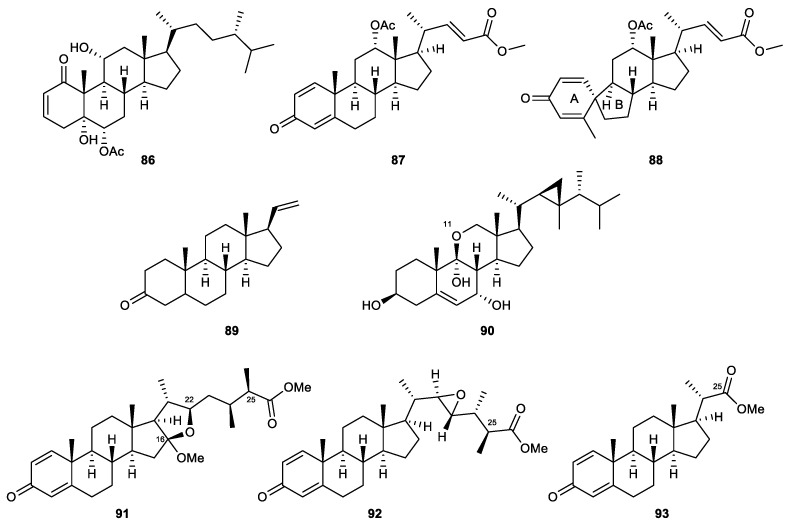
Other steroids (**86**–**93**) isolated from the octocorals.

**Figure 16 marinedrugs-21-00456-f016:**
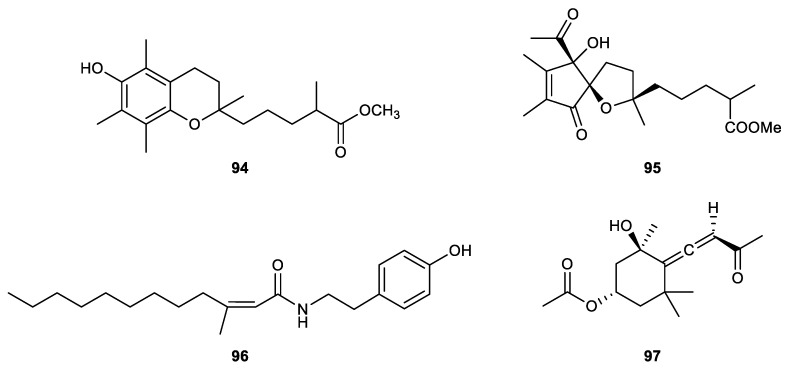
Bioactive metabolites (**94**–**97**) isolated from octocorals.

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
