# Peer review of "Unlocking the Potential of Octocoral-Derived Secondary Metabolites against Neutrophilic Inflammatory Response"

_marinedrugs, 2023, doi:10.3390/md21080456_

Round 1

Reviewer 1 Report

The authors provide a comprehensive review of the knowledge so far on neutrophilic-mediated anti-inflammatory activity of secondary metabolites derived from soft corals. This review accomplishes the goals set by the authors as it summarizes current findings in this area. I believe this review paper is ready for publication after some minor revisions. 

Line 136 - please give examples of neutrophil-targeting anti-inflammatory agents.

Line 144 - Please describe soft corals in more detail. Can they be grown outside their environment in industrial scale? what kind of compounds do they possess? have they been used in the past in traditional medicine? what kind of bioactivities were associated? After the introductory part of inflammation, these soft-coral part is very brief

Line 163 - I  would suggest the authors add a small paragraph summarizing the studies they found in terms of classes of compounds to help guide the readers.

Line 184 - Please describe which diterpene classes are going to be described.

Line 207 - Please check all "in vitro" to ensure italics.

Line 350 -  Please describe which steroid classes are going to be described.

Author Response

Response to Reviewer 1

The authors provide a comprehensive review of the knowledge so far on neutrophilic-mediated anti-inflammatory activity of secondary metabolites derived from soft corals. This review accomplishes the goals set by the authors as it summarizes current findings in this area. I believe this review paper is ready for publication after some minor revisions.

  • We are sincerely grateful for the careful inspections and the valuable comments from the reviewer. We amended all points accordingly.

Line 136 - please give examples of neutrophil-targeting anti-inflammatory agents.

  • Please refer to lines 136-138 (pages 4) in the latest version. The following idea was supplemented as a response for the reviewer’s suggestion: “Among them, several neutrophil-targeting anti-inflammatory agents have been used in clinical stage, such as colchicin, secukinumab, ixekizumab, brodalumab, reparixin, danirixin, sivelestat, and nafamostat”

Line 144 - Please describe soft corals in more detail. Can they be grown outside their environment in industrial scale? what kind of compounds do they possess? have they been used in the past in traditional medicine? what kind of bioactivities were associated? After the introductory part of inflammation, these soft-coral part is very brief

  • Please refer to lines 150-177 (pages 4) in the latest version. The following idea was supplemented as a response for the reviewer’s suggestion: “These marine organisms are named “octocorallia” as they possess eight pinnate tentacles on the oral opening of their polyp tubes, which are used as a tool of food capture. Unlike hard corals, more than 60% of octocoral body is occupied by fleshy part. Therefore, their defensive mechanism against potential predators mainly relies on chemical composition contained in the soft tissues [54,55]. The use of soft corals were recorded in ancient literature as a therapeutic ingredient for the treatment of diarrhea, gastrointestinal bleeding, and neurasthenia [56]. In modern time, a great quantity of studies has been conducted in regards to the chemical constituents of soft corals and their pharmacological potentials. Terpenoids and steroids are reported as chemical compositions commonly discovered in these organisms. Secondary metabolites derived from different species of soft corals showed a diverse range of bioactivities, including cytotoxic, antimicrobial, antimalarial, antifouling, antidiabetic, antixiolytic, antileishmanial, anti-acne, analgesic, antiviral, and anti-inflammatory effects [35-39,45,49-52,57-61]. The promising potentials in such pharmacological effects of soft corals have leveraged them to be hot spots in the race of drug discovery. Aquaculture of soft corals has been conducted since the late 1950s for the purposes of commercialization anreservation. Many techniques such as coral gardening, micro-fragmentation, larval enhancement, and direct transplantation have been developed to meet the demand for mass production of soft corals [38]. Ex situ and in situ are current approaches of coral cultivation. Whereas in situ practice relies on natural environment for the propagation of soft corals, ex situ method produces these marine organisms in controlled conditions. Although ex situ approach is more costly and requires more advanced skills than in situ, it allows optimization of aquaculture conditions and elimination of environmental variablity, so as to facilitate and enhane the biomass and metabolite production of cultivated soft corals. Besides, there is no inteference in the growth of the animals concerning the exposure to deleterious factors such as parasites, competitors, predators, and other hazards [62]. These advantages have driven ex situ practice a suitable and favorable method of aquaculture to serve the drug discovery journey in marine organisms, in particular soft corals.

Line 163 - I would suggest the authors add a small paragraph summarizing the studies they found in terms of classes of compounds to help guide the readers.

  • Please refer to lines 187-194 (pages 5) in the latest version. The following idea was supplemented as a response for the reviewer’s suggestion: “In the current review, the secondary metabolites possessing significant neutrophil-targeting anti-inflammatory effect are classified into sesquiterpenes, diterpenes, biscembranes, steroids, and some miscellaneous compounds. Steroids are the most abundant population with a total of 48 compounds, occupying 49.5% of bioactive agents isolated from octocorals. Ranked in second place, diterpenes comprise of 36 derivatives that are classified into 10 subtypes. The remaining 13.4% of 97 potent isolates include 6 sesquiterpenes, 6 biscembranes, 2 a-tocopherol derivatives, a nitrogen-containing compound, and an allenic norterpenoid ketone.”

Line 184 - Please describe which diterpene classes are going to be described.

  • Please refer to lines 228-231 (page 6) in the latest version. The following idea was supplemented as a response for the reviewer’s suggestion: “Within the scope of the current literature-based investigation, 4 major subtypes of diterpenes, including briarane, cembrane, eunicellin, and xenicane, along with 6 minor diterpene classes such as halimane, verticillane, C-2/C-20-cyclized cembranoid skeleton, norcembranoid, capsosane, and lobane, are included in this part.”

Line 207 - Please check all "in vitro" to ensure italics.

  • The error was corrected.

Line 350 - Please describe which steroid classes are going to be described.

Please refer to lines 402-404 (page 12) in the latest version. The following idea was supplemented as a response for the reviewer’s suggestion: “As per the investigation of related studies, sterols, 9,11-secosterols, gorgostane-type steroids, withanolide-type steroids, steroid glycosides, and and several unclassified steroids were recorded as potential neutrophilic-targeting anti-inflammatory steroids derived from octocorals.”

Reviewer 2 Report

It is well organized manuscript but there are some major concerns.

1. did the author make sure they include all the literature, such as https://doi.org/10.1016/j.molstruc.2022.133995

2. The manuscript has no novelty or interests for reader. So the author should add some thoughtful discussion to give a significant contribution. Such as it would be great if the author could add some MOA information for some compounds and discussion in the manuscript. In addition, is it possible to add some structure activity relationship discussion in the manuscript. 

Some minor suggestion

L146 should be cited as a reference

L529 format

Not sure  We hope in L38 is scientific enough but maybe the review aims to .....

English is good

Author Response

Response to Reviewer 2

It is well organized manuscript but there are some major concerns.

  • We are sincerely grateful for the careful inspections and the valuable comments from the reviewer. We amended all points accordingly.
  1. did the author make sure they include all the literature, such as https://doi.org/10.1016/j.molstruc.2022.133995
  • We would like to focus on the secondary metabolites that exhibited significant inhibitory effects in the in vitro Therefore, some studies will be excluded from our current review as the isolates in these studies only showed moderate to weak effects in the screening tests. Please refer to line 184-187 (page 5). The following idea was supplemented to clarify this point: “A total of 299 compounds isolated from different species of the subclass Octocorallia were screened for their anti-inflammatory potential using the aforementioned in vitro tests. Among them, 97 isolates were considered to exhibit significant inhibitory effects on superoxide anion generation and elastase release, with IC50 equal to or less than 20 mM.”
  1. The manuscript has no novelty or interests for reader. So the author should add some thoughtful discussion to give a significant contribution. Such as it would be great if the author could add some MOA information for some compounds and discussion in the manuscript. In addition, is it possible to add some structure activity relationship discussion in the manuscript.
  • We are grateful for the insight comments from reviewer. We added the MOA information in the text line 655-680 (page 25); the SAR information in the text line 581-651 (page 19-20).

Some minor suggestion L146 should be cited as a reference

  • The error was corrected. Please refer to page 18 in the latest version.

L529 format

  • The error was corrected. Please refer to page 18 in the latest version.

Not sure We hope in L38 is scientific enough but maybe the review aims to .....

  • We would like to focus on the secondary metabolites that exhibited significant inhibitory effects in the in vitro Therefore, some studies will be excluded from our current review as the isolates in these studies only showed moderate to weak effects in the screening tests.

Round 2

Reviewer 2 Report

Thank you for the implement of the manuscript. It is a very comprehensive review. There are still some minor errors but easily to fix. 

Table 1. delete one of 68

L457, (24S) should be  24(S) * S should be Italic. Same for (Z) and N by L570 and check through the manuscript

L827, Chemical Reviews should be Chem. Rev.